# RanDumb: Random Representations Outperform Online Continually Learned Representations

Ameya Prabhu[1]*    Shiven Sinha[2]*    Ponnurangam Kumaraguru[2]    Philip H.S. Torr[1]

Ozan Sener[3]+    Puneet K. Dokania[1]+

[1]University of Oxford     [2]IIIT Hyderabad     [3]Apple

https://github.com/drimpossible/RanDumb

## Abstract

Continual learning has primarily focused on the issue of catastrophic forgetting and the associated stability-plasticity tradeoffs. However, little attention has been paid to the efficacy of continually learned representations, as representations are learned alongside classifiers throughout the learning process. Our primary contribution is empirically demonstrating that existing online continually trained deep networks produce inferior representations compared to a simple pre-defined random transforms. Our approach projects raw pixels using a fixed random transform, approximating an RBF-Kernel initialized before any data is seen. We then train a simple linear classifier on top without storing any exemplars, processing one sample at a time in an online continual learning setting. This method, called RanDumb, significantly outperforms state-of-the-art continually learned representations across all standard online continual learning benchmarks. Our study reveals the significant limitations of representation learning, particularly in low-exemplar and online continual learning scenarios. Extending our investigation to popular exemplar-free scenarios with pretrained models, we find that training only a linear classifier on top of pretrained representations surpasses most continual fine-tuning and prompt-tuning strategies. Overall, our investigation challenges the prevailing assumptions about effective representation learning in online continual learning.

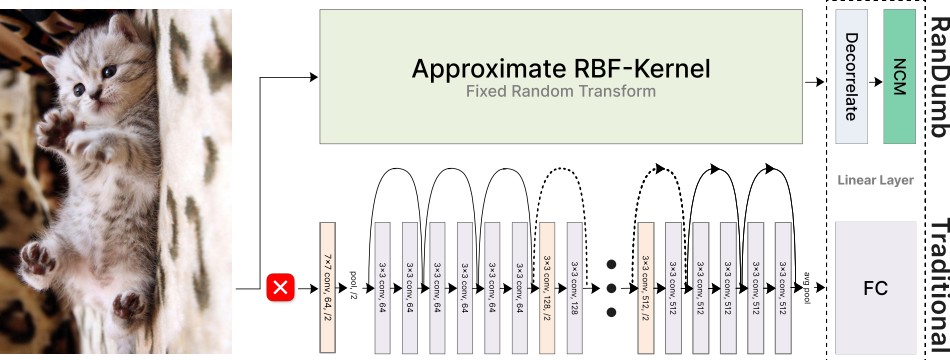

Figure 1: Our primary analysis in this work is ablating the deep feature extractor (bottom center) by replacing it with a random projection (top center) to isolate the effect of online continual representation learning in the deep network. We demonstrate that random projections not only match but consistently outperform continually learned representations, highlighting the poor quality of the continually learned representations. RanDumb (top) maps raw pixels to a high-dimensional space using random Fourier projections, then decorrelates the features using the Mahalanobis distance [43] and classifies based on the nearest class mean.

---

*authors contributed equally, + equal advising

38th Conference on Neural Information Processing Systems (NeurIPS 2024).

# 1 Introduction

Continual learning aims to develop models capable of learning from non-stationary data streams, inspired by the lifelong learning abilities exhibited by humans and the prevalence of such real-world applications (see Verwimp et al. [65] for a survey). It is characterized by sequentially arriving tasks, coupled with additional computational and memory constraints [33, 38, 54, 63, 49].

Building on the foundations of supervised deep learning, the prevalent approach in continual learning has been to jointly train representations alongside classifiers. This approach simply follows from the assumption that learned representations are expected to outperform fixed representation functions such as kernel classifiers, as demonstrated in supervised deep learning [34, 23, 57]. However, this assumption is never validated in continual learning, with scenarios having limited updates where networks might not be trained until convergence, such as online continual learning (OCL).

In this paper, we study the efficacy of representations derived from continual learning algorithms. Surprisingly, our findings suggest that these representations might not be as beneficial as presumed. To test this, we introduce a simple baseline method named RanDumb, which combines a random representation function with a straightforward linear classifier, illustrated in detail in Figure 1. Our empirical evaluations, summarized in Table 1 (left, top), reveal that despite replacing the representation learning with a pre-defined random representation, RanDumb surpasses current state-of-the-art methods in latest online continual learning benchmarks [76].

We further expand our evaluations to scenarios incorporating methods that use pre-trained feature extractors [68]. By substituting our random projections with these feature extractors and retaining the linear classifier, RanDumb again outperforms leading methods as shown in Table 1 (right, top).

## 1.1 Technical Summary: Construction of RanDumb and Empirical Findings

*Design.* RanDumb first projects input pixels into a high-dimensional space using a fixed kernel based on random Fourier basis, which is a low-rank data-independent approximation of the RBF Kernel [52]. Then, we use a simple linear classifier which first normalizes distances across different feature dimensions (anisotropy) with Mahalanobis distance [43] and then uses nearest class means for classification [44]. In scenarios with pretrained feature extractors, we use the fixed pretrained model as embedder and learn a linear classifier as described above, similar to Hayes and Kanan [27].

*Key Properties.* RanDumb needs no storage of exemplars and requires only one pass over the data in a one-sample-per-timestep fashion. Furthermore, it only requires online estimation of the sample covariance matrix and nearest class mean.

**Key Finding 1:** *Poor Representation Learning.* We compare RanDumb with leading methods: VAE-GC [64] in Table 1 (left, middle) and SLCA [79] in Table 1 (right, middle). The primary distinction between them is their representation: RanDumb uses a fixed function (random/pretrained network),

Table 1: **(Left) Online Continual Learning.** Performance comparison of RanDumb on the PEC setup [76] and VAE-GC [64]. Setup and numbers borrowed from PEC [76]. RanDumb outperforms the best OCL method. **(Right) Offline Continual Learning.** Performance comparison with ImageNet21K ViT-B16 model using 2 initial classes and 1 new class per task. RanPAC-imp is an improved version of the RanPAC code which mitigates the instability issues in RanPAC. RanDumb nearly matches performance of joint for both online and offline, demonstrating the inefficacy of current benchmarks.

| Method | MNIST | CIFAR10 | CIFAR100 | m-IMN | Method | CIFAR | IN-A | IN-R | CUB | OB | VTAB | Cars |
|---|---|---|---|---|---|---|---|---|---|---|---|---|
| Comparison with Best Method | | | | | Comparison with Best Method | | | | | | | |
| Best (PEC) | 92.3 | 58.9 | 26.5 | 14.9 | Best (RanPAC-imp) | 89.4 | 33.8 | 69.4 | 89.6 | 75.3 | 91.9 | 57.3 |
| RanDumb (Ours) | 98.3 | 55.6 | 28.6 | 17.7 | RanDumb (Ours) | 86.8 | 42.2 | 64.9 | 88.5 | 75.3 | 92.4 | 67.1 |
| Improvement | +6.0 | -3.3 | +2.1 | +2.8 | Improvement | -2.6 | +8.4 | -4.5 | -1.1 | +0.0 | +0.5 | +9.8 |
| Random vs. Learned Representations | | | | | Random vs. Finetuned Representations | | | | | | | |
| VAE-GC | 84.0 | 42.7 | 19.7 | 12.1 | SLCA | 86.8 | - | 54.2 | 82.1 | - | - | 18.2 |
| RanDumb (Ours) | 98.3 | 55.6 | 28.6 | 17.7 | RanDumb (Ours) | 86.8 | 42.2 | 64.9 | 88.5 | 75.3 | 92.4 | 67.1 |
| Improvement | +14.3 | +12.9 | +8.9 | +5.6 | Improvement | +0.0 | - | +10.7 | +6.4 | - | - | +48.9 |
| Scope of Improvement | | | | | Scope of Improvement | | | | | | | |
| Joint (One Pass) | 98.3 | 74.2 | 33.0 | 25.3 | Joint | 93.8 | 70.8 | 86.6 | 91.1 | 83.8 | 95.5 | 86.9 |
| RanDumb (Ours) | 98.3 | 55.6 | 28.6 | 17.7 | RanDumb (Ours) | 86.8 | 42.2 | 64.9 | 88.5 | 75.3 | 92.4 | 67.1 |
| Gap Covered. (%) | 100% | 75% | 87% | 70% | Gap Covered. (%) | 93% | 60% | 75% | 97% | 92% | 97% | 77% |

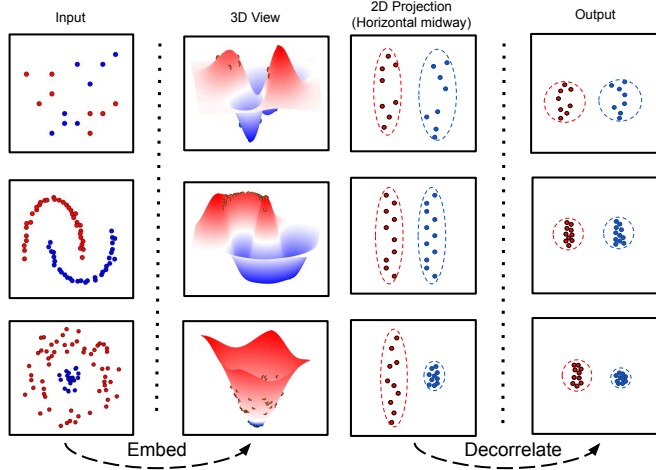

| Input | 3D View | 2D Projection (Horizontal midway) | Output |

Figure 2: RanDumb projects the datapoints to a high-dimensional space to create a clearer separation between classes. Subsequently, it corrects the anisotropy across feature dimensions, scaling them to be unit variance each. This allows cosine similarity to accurately separates classes. The figure is adapted from [48].

whereas VAE-GC and SLCA further continually trained deep networks. RanDumb consistently surpasses VAE-GC and SLCA by wide margins of 5-15%. This shows that state-of-the-art online continual learning algorithms fail to learn effective representations across standard exemplar-free continual learning benchmarks.

**Finding 2: *Over-Constrained Benchmarks.*** Given the demonstrated limitations of existing continual representation learning methods, an important question arises: Can better methods learn more effective representations? To explore this, we evaluated the performance of RanDumb against joint training, models trained without continual learning constraints, in both online and offline settings, as shown in Table 1 (left, bottom) and Table 1 (right, bottom). Our straightforward baseline, RanDumb, bridges 70-90% of the performance gap relative to the respective joint classifiers in both scenarios. This significant recovery of performance by such a simple method suggests that if our goal is to advance the study of representation learning, current benchmarks may be overly restrictive and not conducive to truly effective representation learning.

We highlight that the goal in our work is not to introduce a state-of-the-art continual learning method, but challenge prevailing assumptions and open a discussion on the efficacy of representation learning in continual learning algorithms, especially in online and low-exemplar scenarios.

## 2 RanDumb: Mechanism & Intuitions

RanDumb has two main elements: random projection and the dumb learner. We illustrate the mechanism of RanDumb using three toy examples in Figure 2. To classify a test sample $\mathbf{x}_{\text{test}}$, we start with a simple classifier, the nearest class mean (NCM). It predicts the class among $C$ classes by highest value of the similarity function $f$ among class means $\mu_i$:

$$y_{\text{pred}} = \underset{i \in \{1, \ldots, |C|\}}{\arg \max} f(\mathbf{x}_{\text{test}}, \mu_i), \quad \text{where} \quad f(\mathbf{x}_{\text{test}}, \mu_i) := \mathbf{x}_{\text{test}}^{\top} \mu_i \tag{1}$$

and $\mu_i$ are the class-means in the pixel space: $\mu_i = \frac{1}{|C_i|} \sum_{\mathbf{x} \in C_i} \mathbf{x}$. RanDumb adds two additional components to this classifier: 1) Kernelization and 2) Decorrelation.

**Kernelization:** Classes are typically not linearly separable in the pixel space, unlike in the feature space of deep models. Hence, we randomly project the pixels into a high-dimensional representation space, computing all distances between the data and class-means in this embedding space. This phenomena is illustrated on three toy examples to build intuitions in Figure 2 (Embed). We use an RBF-Kernel, which for two points $\mathbf{x}$ and $\mathbf{y}$ is defined as: $K_{\text{RBF}}(\mathbf{x}, \mathbf{y}) = \exp\left(-\gamma \|\mathbf{x} - \mathbf{y}\|^2\right)$ where $\gamma$ is a scaling parameter. However, calculating the RBF kernel is not possible due to the online continual learning constraints preventing computation of pairwise-distance between all points. We use a data-independent approximation, random Fourier projection $\phi(\mathbf{x})$, as given in [52]:

$$K_{\text{RBF}}(\mathbf{x}, \mathbf{y}) \approx \phi(\mathbf{x})^T \phi(\mathbf{y})$$

where the random Fourier features $\phi(\mathbf{x})$ are defined by first sampling $D$ vectors $\{\omega_1, \ldots, \omega_D\}$ from a Gaussian distribution with mean zero and covariance matrix $2\gamma\mathbf{I}$, where $\mathbf{I}$ is the identity matrix. Then $\phi(\mathbf{x})$ is a $2D$-dimensional feature, defined as:

$$\phi(\mathbf{x}) = \frac{1}{\sqrt{D}} \left[ \cos(\omega_1^T \mathbf{x}), \sin(\omega_1^T \mathbf{x}), .., \cos(\omega_D^T \mathbf{x}), \sin(\omega_D^T \mathbf{x}) \right]$$

We keep these $\omega$ bases fixed throughout online learning. Thus, we obtain our modified similarity function from Equation 1 as:

$$f(\mathbf{x}_{\text{test}}, \mu_i) := \phi(\mathbf{x}_{\text{test}})^\top \bar{\mu}_i \tag{2}$$

where $\bar{\mu}_i$ are the class-means in the kernel space:

$$\bar{\mu}_i = \frac{1}{|C_i|} \sum_{\mathbf{x} \in C_i} \phi(\mathbf{x})$$

**Decorrelation:** Projected raw pixels have feature dimensions with different variances (anisotropic). Hence, instead of naively computing $\phi(\mathbf{x}_{\text{test}})^\top \bar{\mu}_i$, we further decorrelate the feature dimensions using a Mahalonobis distance with the shrinked covariance matrix $\mathbf{S}$ using OAS shrinkage [15], inverse obtained by least squares minimization $(\mathbf{S} + \lambda\mathbf{I})$. We illustrate this phenomena as well on three toy examples in Figure 2 (Decorrelate) to build intuitions. Our similarity function finally is:

$$f(\mathbf{x}_{\text{test}}, \mu_i) := (\phi(\mathbf{x}_{\text{test}}) - \bar{\mu}_i)^T \mathbf{S}^{-1} (\phi(\mathbf{x}_{\text{test}}) - \bar{\mu}_i) \tag{3}$$

***Online Computation.*** Our random projection is fixed before seeing any data. During continual learning, we only perform online update on the running class mean and empirical covariance matrix[2].

## 3   Experiments

We compare RanDumb with algorithms across online continual learning benchmarks with an emphasis on exemplar-free and low-exemplar storage regime. All numbers in tables with the caption (Ref: table and citation) except our method are taken from the aforementioned table in the cited paper.

***Benchmarks.*** The benchmarks which we used in our experiments are summized in Table on the right. We aim for a comprehensive coverage and show results on four standard online continual learning benchmarks (A, B, D, E) which reflect the latest trends ('22-'24) across exemplar-free, contrastive-training[3], meta-continual learning, and network-expansion based approaches re-

| Setup | Num Passes | #Classes Per Task | #Samples Per Step | #Stored Exemplars | Contrastive Augment |
|---|---|---|---|---|---|
| Method: RanDumb | 1 | 1 | 1 | 0 | No |
| A (Zając et al. [76]) | 1 | 1 | 10 | 0 | No |
| B1 (Guo et al. [25]) | 1 | 2 | 10 | 100-2000 | No |
| B2 (Guo et al. [25]) | 1 | 2 | 10 | 100-1000 | Yes |
| C (Smith et al. [60]) | Many | 10 | All | 0 | No |
| D (Wu et al. [71]) | 1 | 2-10 | 10 | 1000 | No |
| E (Ye and Bors [75]) | 1 | 2-5 | 10 | 1000-5000 | No |
| F (Wang et al. [68], modified) | Many | 1 | All | 0 | No |

spectively. We also evaluate on a rehearsal-free offline continual learning benchmark C. These benchmarks are ordered by increasingly relaxed constraints, moving further away from the training scenario of RanDumb. Benchmark A closely matches RanDumb with one class per timestep and no stored exemplars. Benchmark B, D, E progressively relax the constraints on exemplars and classes per timestep. Benchmark C and E remove the online constraint by allowing unrestricted training and sample access within a task without exemplar-storage of past tasks. Benchmark F allows using large pretrained models, modified by us with one class per task, i.e. testing learning over longer timespans.

We further test on exemplar-free scenarios in offline continual learning using Benchmark F [68] with the challenging one-class per task constraint borrowed from [76]. This benchmark allows using pretrained models along with unrestricted training time and access to all class samples at each timestep. However, RanDumb is restricted to learning from a single pass seeing only one sample at a time. RanDumb only learns a linear classifier over a given pretrained model in Benchmark F.

---

[2]Online update for the inverse of the covariance matrix is possible using the Sherman–Morrison formula.

[3]Benchmark B is split into two sections: (B1) methods that do not rely on contrastive learning and heavy augmentation, and (B2) approaches that incorporate contrastive learning and extra augmentations.

Table 2: **Benchmark A** *(Ref: Table 1 from PEC [76]).* We compare RanDumb in a 1-class per task setting referred as 'Dataset (num_tasks/1)'. We observe that RanDumb outperforms all approaches across all datasets by 2-6% margins, with an exception of latest work PEC [76] on CIFAR10.

| | Method | Memory | MNIST (10/1) | CIFAR-10 (10/1) | CIFAR-100 (100/1) | miniImageNet (100/1) |
|---|---|---|---|---|---|---|
| | *Fine-tuning* | all | 10.1± 0.0 | 10.0± 0.0 | 1.0± 0.0 | 1.0± 0.0 |
| | *Joint, 1 epoch* | all | 98.3± 0.0 | 74.2± 0.1 | 33.0± 0.2 | 25.3± 0.2 |
| | ER [13] | 500 | 84.4± 0.3 | 40.6± 1.1 | 12.5± 0.3 | 5.7± 0.2 |
| | A-GEM [12] | 500 | 59.8± 0.8 | 10.2± 0.1 | 1.0± 0.0 | 1.1± 0.1 |
| | iCaRL [54] | 500 | 83.1± 0.3 | 37.8± 0.4 | 5.7± 0.1 | 7.5± 0.1 |
| *Rehearsal* | BiC [72] | 500 | 86.0± 0.4 | 35.9± 0.4 | 6.4± 0.3 | 1.5± 0.1 |
| *Based* | ER-ACE [10] | 500 | 87.8±0.2 | 39.9±0.5 | 8.2±0.2 | 5.7±0.2 |
| *Methods* | DER [9] | 500 | 91.7± 0.1 | 40.0± 1.5 | 1.0± 0.1 | 1.0± 0.0 |
| | DER++ [9] | 500 | 91.9± 0.2 | 35.6± 2.4 | 6.2± 0.4 | 1.4± 0.1 |
| | X-DER [8] | 500 | 83.0± 0.1 | 43.2± 0.5 | 15.6± 0.1 | 8.2± 0.4 |
| | GDumb [49] | 500 | 91.0±0.2 | 50.7±0.7 | 8.2±0.2 | - |
| | EWC [33] | 0 | 10.1± 0.0 | 10.6± 0.4 | 1.0± 0.0 | 1.0± 0.0 |
| | SI [77]) | 0 | 12.7± 1.0 | 10.1± 0.1 | 1.1±0.0 | 1.0±0.1 |
| | LwF [37] | 0 | 11.8 ± 0.6 | 10.1± 0.1 | 0.9±0.0 | 1.0± 0.0 |
| *Rehearsal* | LT [78] | 0 | 10.9± 0.9 | 10.0± 0.2 | 1.1± 0.1 | 1.0± 0.0 |
| *Free* | Gen-NCM [31] | 0 | 82.0± 0.0 | 27.7± 0.0 | 10.0± 0.0 | 7.5± 0.0 |
| *Methods* | Gen-SLDA [27] | 0 | 88.0± 0.0 | 41.4± 0.0 | 18.8± 0.0 | 12.9± 0.0 |
| | VAE-GC [64] | 0 | 84.0± 0.5 | 42.7± 1.3 | 19.7± 0.1 | 12.1± 0.1 |
| | PEC [76] | 0 | 92.3± 0.1 | **58.9± 0.1** | 26.5± 0.1 | 14.9± 0.1 |
| | RanDumb (Ours) | 0 | **98.3** (+5.9) | 55.6 (-3.3) | **28.6** (+2.1) | **17.7** (+2.8) |

Table 3: **Benchmark B.1** *(Ref: Table adopted from OnPro [69], OCM[25])* We compare RanDumb in many-classes per task setting referred as 'Dataset (num_tasks/num_classes_per_task)'. We categorize memory buffer sizes with 'M'. RanDumb outperforms the competing approaches without heavy-augmentations by 3-20% margins despite being exemplar free. Only in one case, it is second best.

| Method | MNIST (5/2) $M = 0.1k$ | CIFAR10 (5/2) $M = 0.1k$ | $M = 0.2k$ | CIFAR100 (10/10) $M = 0.5k$ | $M = 1k$ | CIFAR100 (50/2) $M = 1k$ | TinyImageNet (100/2) $M = 1k$ | $M = 2k$ |
|---|---|---|---|---|---|---|---|---|
| AGEM [12] | 56.9±5.2 | 17.7±0.3 | 22.7±1.8 | 5.8±0.2 | 5.9±0.1 | 1.8±0.2 | 0.8±0.1 | 0.9±0.1 |
| GSS [4] | 70.4±1.5 | 18.4±0.2 | 26.9±1.2 | 8.1±0.2 | 11.1±0.2 | 4.3±0.2 | 1.1±0.1 | 3.3±0.5 |
| ER [13] | 78.7±0.4 | 19.4±0.6 | 29.7±1.0 | 8.7±0.3 | 15.7±0.3 | 8.3±0.3 | 1.2±0.1 | 5.6±0.5 |
| ASER [58] | 61.6±2.1 | 20.0±1.0 | 27.8±1.0 | 11.0±0.3 | 16.4±0.3 | 9.6±1.3 | 2.2±0.1 | 5.3±0.3 |
| MIR [3] | 79.0±0.5 | 20.7±0.7 | 37.3±0.3 | 9.7±0.3 | 15.7±0.2 | 12.7±0.3 | 1.4±0.1 | 6.1±0.5 |
| ER-AML [10] | 76.5±0.1 | - | 40.5±0.7 | - | 16.1±0.4 | - | - | 5.4±0.2 |
| iCaRL [54] | - | 31.0±1.2 | 33.9±0.9 | 12.8±0.4 | 16.5±0.4 | - | 5.0±0.3 | 6.6±0.4 |
| DER++ [9] | 74.4±1.1 | 31.5±2.9 | 44.2±1.1 | 16.0±0.6 | 21.4±0.9 | 9.3±0.3 | 3.7±0.4 | 5.1±0.8 |
| GDumb [49] | 81.2±0.5 | 23.3±1.3 | 35.9±1.1 | 8.2±0.2 | 18.1±0.3 | 18.1±0.3 | 4.6±0.3 | **12.6±0.1** |
| CoPE [19] | - | 33.5±3.2 | 37.3±2.2 | 11.6±0.4 | 14.6±1.3 | - | 2.1±0.3 | 2.3±0.4 |
| DVC [25] | - | 35.2±1.7 | 41.6±2.7 | 15.4±0.3 | 20.3±1.0 | - | 4.9±0.6 | 7.5±0.5 |
| Co²L [11] | 83.1±0.1 | - | 42.1±1.2 | - | 17.1±0.4 | - | - | 10.1±0.2 |
| R-RT [6] | 89.1±0.3 | - | 45.2±0.4 | - | 15.4±0.3 | - | - | 6.6±0.3 |
| CCIL [46] | 86.4±0.1 | - | 50.5±0.2 | - | 18.5±0.3 | - | - | 5.6±0.9 |
| IL2A [82] | 90.2±0.1 | - | 54.7±0.5 | - | 18.2±1.2 | - | - | 5.5±0.7 |
| BiC [72] | 90.4±0.1 | - | 48.2±0.7 | - | 21.2±0.3 | - | - | 10.2±0.9 |
| SSIL [1] | 88.2±0.1 | - | 49.5±0.2 | - | 26.0±0.1 | - | - | 9.6±0.7 |
| *Rehearsal-Free* | | | | | | | | |
| PASS [82] | - | 33.7±2.2 | 33.7±2.2 | 7.5±0.7 | 7.5±0.7 | - | 0.5±0.1 | 0.5±0.1 |
| RanDumb (Ours) | **98.3** (+7.8) | **55.6** (+20.4) | **55.6** (+5.9) | **28.6** (+12.6) | **28.6** (+2.6) | **28.6** (+10.5) | **11.6** (+6.6) | 11.6 (-1.0) |

We use LAMDA-PILOT [61] codebase for all methods, except RanPAC and SLDA for which use their codebases. We use the original hyperparameters. We only change initial classes to 2 and number of classes per task to 1 and test using both ImageNet21K and ImageNet1K ViT-B/16 models.

***Implementation Details (RanDumb).*** We evaluate RanDumb using five datasets: MNIST, CIFAR10, CIFAR100, TinyImageNet200, and miniImageNet100. For the latter two, we downscale all images to 32x32. We augment each datapoint with flipped version, hence two images are seen by the classifier at each timestep (except for MNIST and Benchmark F). We normalize all images and flatten them into vectors, obtaining 784-dim input vectors for MNIST and 3072-dim input vectors for all the other. For Benchmark F, we compare RanDumb on seven datasets used in LAMDA-PILOT, replacing ObjectNet

Table 4: **(Left) Benchmark B.2** *(Ref: Table from OnPro [69])* We compare RanDumb with contrastive representation learning based approaches which additionally use sophisticated augmentations. We observe that RanDumb often outperforms these sophisticated methods despite all of these factors on small-exemplar settings. **(Right) Benchmark C** *(Ref: Table 2 from [60])*. We compare RanDumb with latest rehearsal-free methods. RanDumb outperforms them by 4% margin.

| Method | MNIST (5/2) $M = 0.1k$ | CIFAR10 (5/2) $M = 0.1k$ | CIFAR100 (10/10) $M = 0.5k$ | TinyImageNet (100/2) $M = 1k$ |
|---|---|---|---|---|
| SCR [40] | 86.2±0.5 | 40.2±1.3 | 19.3±0.6 | 8.9±0.3 |
| OCM [25] | 90.7±0.1 | 47.5±1.7 | 19.7±0.5 | 10.8±0.4 |
| OnPro [69] | - | **57.8±1.1** | 22.7±0.7 | **11.9±0.3** |
| *Rehearsal-Free* | | | | |
| RanDumb | **98.3** (+7.5) | 55.6 (-2.2) | **28.6** (+5.9) | 11.6 (-0.3) |

| Method | CIFAR100 (10/10) |
|---|---|
| *Rehearsal-Free* | |
| PredKD [37] | 24.6 |
| PredKD + FeatKD | 12.4 |
| PredKD + EWC | 23.3 |
| PredKD + L2 | 21.5 |
| RanDumb (Ours) | **28.6** (+4.0) |

with Stanford Cars as ObjectNet license prohibits training models. We use the 768-dimensional features from the same pretrained ViT-B models used in this benchmark. We measure accuracy on the test set of all past seen classes after completing the full one-pass. We take the average accuracy after the last task on all past tasks [76, 25, 68]. In Benchmark A and F, since we have one class per task, the average accuracy across past tasks is the same regardless of the task ordering. In Benchmarks A-E, all datasets have the same number of samples, hence similarly the average accuracy across past tasks is the same regardless of the task ordering. We used the Scikit-Learn implementation of Random Fourier Features [52] with 25K embedding size, $\gamma = 1.0$. We use progressively increasing ridge regression parameter ($\lambda$) with dataset complexity, $\lambda = 10^{-6}$ for MNIST, $\lambda = 10^{-5}$ for CIFAR10/100 and $\lambda = 10^{-4}$ for TinyImageNet200/miniImageNet100.

## 3.1 Results

**Benchmark A (single-class per task).** We assess continual learning models in the challenging setup of one class per timestep, closely mirroring our training assumptions, and present our results in Table 2. Comparing across rows, and see that RanDumb improves over prior state-of-the-art across all datasets with 2-6% margins. The only exception is PEC on CIFAR10, where RanDumb underperforms by 3.3%. Nonetheless, it outperforms the second-best model, GDumb with a 500 memory size, by 4.9%.

**Benchmark B.1 (many-classes per task).** We present our results comparing with non-contrastive methods in Table 3. We notice that scenario allows two classes per task and relaxes the memory constraints for online continual learning methods, allowing for higher accuracies compared to Benchmark A. Despite that, RanDumb outperforms latest OCL algorithms on MNIST, CIFAR10 and CIFAR100—often by margins exceeding 10%. The lone exception is GDumb achieving a higher performance with 2K memory samples on TinyImageNet, indicating that this already is in the high-memory regime.

**Benchmark B.2 (many-classes per task, with contrastive losses and data augmentations).** We additionally compare our performance with the latest OCL approaches using contrastive losses with sophisticated data augmentations. As shown in in Table 4 (Left), these advancements provide large performance improvements over methods from Benchmark B.1. To compensate, we compare on lower exemplar budgets. The best approach, OnPro [69], outperforms RanDumb on CIFAR10 by 2.2% and TinyImageNet by 0.3%, but falls significantly short on CIFAR100 by 5.9%. Overall, RanDumb achieves strong results compared to representation learning using state-of-the-art contrastive learning approaches customized to continual learning, despite storing no exemplars.

**Benchmark C (rehearsal-free).** We compare against offline rehearsal-free continual learning approaches in Table 4 (Right) on CIFAR100. Despite online training, RanDumb outperforms PredKD by over 4% margins.

**Benchmark D (meta-continual learning).** We compare performance of RanDumb against meta-continual learning methods, which require large exemplars with buffer sizes of 1K in Table 5 (left). RanDumb achieves strong performance under these conditions, exceeding all prior work by a large margin of 9.1% on CIFAR100 and outperforms all but VR-MCL approach on the TinyImageNet dataset. GDumb performs the best on CIFAR10, indicating this is already in a large-exemplar regime uniquely unsuited for RanDumb.

Table 5: **(Left) Benchmark D** *(Ref: Table 2 from VR-MCL [71])* We compare RanDumb with meta-continual learning approaches operating in a high memory setting, allowing buffer sizes up to 1K exemplars. RanDumb outperforms all methods except VR-MCL on TinyImageNet. RanDumb also surpasses all prior work by a substantial 9.1% on CIFAR100. Allowing generous replay buffers shifts scenarios to a high exemplar regime where GDumb performs the best on CIFAR10. Yet RanDumb competes favorably even under these conditions. **(Right) Benchmark E** *(Ref: Table 1 from SEDEM [75])* We compare RanDumb with network expansion based approaches. Despite allowing access to much larger memory buffers, RanDumb matches the performance of best method SEDEM on MNIST, while exceeding it by 0.3% on CIFAR10 and 3.8% on CIFAR100.

| Method | CIFAR10 (5/2) $M = 1k$ | CIFAR100 (10/10) $M = 1k$ | TinyImageNet (20/10) $M = 1k$ |
|---|---|---|---|
| Finetune | $17.0 \pm 0.6$ | $5.3 \pm 0.3$ | $3.9 \pm 0.2$ |
| LWF [37] | $18.8 \pm 0.1$ | $5.6 \pm 0.4$ | $4.0 \pm 0.3$ |
| A-GEM [12] | $18.4 \pm 0.2$ | $6.0 \pm 0.2$ | $4.0 \pm 0.2$ |
| IS [77] | $17.4 \pm 0.2$ | $5.2 \pm 0.2$ | $3.3 \pm 0.3$ |
| MER [55] | $36.9 \pm 2.4$ | – | – |
| La-MAML [26] | $33.4 \pm 1.2$ | $11.8 \pm 0.6$ | $6.74 \pm 0.4$ |
| GDumb [49] | $\mathbf{61.2 \pm 1.0}$ | $18.1 \pm 0.3$ | $4.6 \pm 0.3$ |
| ER [13] | $43.8 \pm 4.8$ | $16.1 \pm 0.9$ | $11.1 \pm 0.4$ |
| DER [9] | $29.9 \pm 2.9$ | $6.1 \pm 0.1$ | $4.1 \pm 0.1$ |
| DER++ [9] | $52.3 \pm 1.9$ | $11.8 \pm 0.7$ | $8.3 \pm 0.3$ |
| CLSER [5] | $52.8 \pm 1.7$ | $17.9 \pm 0.7$ | $11.1 \pm 0.2$ |
| OCM [25] | $53.4 \pm 1.0$ | $14.4 \pm 0.8$ | $4.5 \pm 0.5$ |
| ER-OBC [18] | $54.8 \pm 2.2$ | $17.2 \pm 0.9$ | $11.5 \pm 0.2$ |
| VR-MCL [71] | $56.5 \pm 1.8$ | $19.5 \pm 0.7$ | $\mathbf{13.3 \pm 0.4}$ |
| *Rehearsal-Free* | | | |
| RanDumb (Ours) | 55.6 (-5.6) | **28.6** (+9.1) | 11.6 (-1.7) |

| Method | MNIST (5/2) $M = 2k$ | CIFAR10 (5/2) $M = 1k$ | CIFAR100 (20/5) $M = 5k$ |
|---|---|---|---|
| Finetune | $19.8 \pm 0.1$ | $18.5 \pm 0.3$ | $3.5 \pm 0.1$ |
| MIR [3] | $93.2 \pm 0.4$ | $42.8 \pm 2.2$ | $20.0 \pm 0.6$ |
| GEM [12] | $93.2 \pm 0.4$ | $24.1 \pm 2.5$ | $11.1 \pm 2.4$ |
| iCARL [54] | $83.9 \pm 0.2$ | $37.3 \pm 2.7$ | $10.8 \pm 0.4$ |
| G-MED [32] | $82.2 \pm 2.9$ | $47.5 \pm 3.2$ | $19.6 \pm 1.5$ |
| GSS [4] | $92.5 \pm 0.9$ | $38.5 \pm 1.4$ | $13.1 \pm 0.9$ |
| CoPE [19] | $93.9 \pm 0.2$ | $48.9 \pm 1.3$ | $21.6 \pm 0.7$ |
| CURL [53] | $92.6 \pm 0.7$ | - | - |
| CNDPM [36] | $95.4 \pm 0.2$ | $48.8 \pm 0.3$ | $22.5 \pm 1.3$ |
| Dynamic-OCM [74] | $94.0 \pm 0.2$ | $49.2 \pm 1.5$ | $21.8 \pm 0.7$ |
| SEDEM [75] | $98.3 \pm 0.2$ | $55.3 \pm 1.3$ | $24.8 \pm 1.2$ |
| *Rehearsal-Free* | | | |
| RanDumb (Ours) | **98.3** (0.0) | **55.6** (+0.3) | **28.6** (+3.8) |

Figure 3: Accuracy of RanDumb with respect to embedding dimensionality across datasets.

**Benchmark E (network-expansion).** We compare RanDumb against network expansion-based online continual learning methods in Table 5 (right). These approaches grow model capacity to mitigate forgetting while dealing with shifts in the data distribution, and are allowed larger memory buffers. RanDumb matches the performance of the state-of-the-art method SEDEM [75] on MNIST, while exceeding it by 0.3% on CIFAR10 and 3.8% on CIFAR100.

## 3.2 Analysis of RanDumb

**Ablating Components of RanDumb.** We ablate the contribution of only using Random Fourier features for embedding and decorrelation to the overall performance of RanDumb in Table 6 (left, top). Ablating the decorrelation and relying solely on random Fourier features, colloquially dubbed Kernel-NCM, has performance drops ranging from 6-25% across the datasets. Replacing random Fourier features with raw features, *ie.* the SLDA baseline, leads to pronounced drop in performance ranging from 3-14% across the datasets. Moreover, ablating both components results in the base nearest class mean classifier, and exhibits the poorest performance with an average reduction of 17%. Therefore, both decorrelation and random embedding are crucial for RanDumb.

**Impact of Embedding Dimensions.** We vary the dimensions of the random Fourier features ranging from compressing 3K input dimensions to 1K to projecting it to 25K dimensions and evaluate its impact on performance in Figure 3. Surprisingly, the random projection to a 3x compressed 1K dimensional space allows for significant performance improvement over not using embedding, given in Table 6 (left, top). Furthermore, increasing the dimension from 1K to 25K results in improvements of 3.6%, 10.4%, 7.0%, and 2.5% on MNIST, CIFAR10, CIFAR100, and TinyImageNet respectively. Increasing the embedding sizes beyond 15K, however, only results in modest improvements of 0.1%, 1.4%, 1.1% and 0.2% on the same datasets, indicating 15K dimensions would be a good point for a performance-computational cost tradeoff.

Table 6: **(Left) Analysis of RanDumb**: We study contributions of decorrelation, random embedding, and data augmentation. We further vary the embedding sizes and regularisation parameter. Finally, we compare with alternate embeddings. (Right) **Architectures** *(Ref: Table 1 from Mirzadeh et al. [45])* RanDumb surpasses continual representation learning across a wide range of architectures, achieving close to 94% of the joint performance.

| Method | MNIST (10/1) | CIFAR10 (10/1) | CIFAR100 (10/1) | T-ImNet (200/1) | m-ImNet (100/1) |
|---|---|---|---|---|---|
| *Ablating Components of RanDumb* | | | | | |
| RanDumb | 98.3 | 55.6 | 28.6 | 11.1 | 17.7 |
| -Decorrelate | 83.8 (-14.5) | 30.0 (-25.6) | 12.0 (-16.6) | 4.7 (-6.4) | 8.9 (-8.8) |
| -Embed | 88.0 (-10.3) | 41.6 (-14.0) | 19.0 (-9.6) | 8.0 (-3.1) | 12.9 (-4.8) |
| -Both | 82.1 (-16.2) | 28.5 (-27.1) | 10.4 (-18.2) | 4.1 (-7.0) | 7.28 (-10.4) |
| *Effect of Adding Flip Augmentation* | | | | | |
| With | - | 55.6 | 28.6 | 11.1 | 17.7 |
| Without | 98.3 | 52.5 (-3.1) | 26.9 (-1.7) | 10.7 (-0.4) | 16.6 (-1.1) |
| *Variation with Ridge Parameter $\lambda$* | | | | | |
| $\lambda = 10^{-6}$ | 98.3 | 53.9 | 27.8 | 10.3 | 15.8 |
| $\lambda = 10^{-5}$ | - | 55.6 | 28.6 | 11.1 | 15.9 |
| $\lambda = 10^{-4}$ | 96.6 | 52.6 | 26.1 | 11.6 | 17.7 |
| *Variation Across Embedding Projections* | | | | | |
| No-Embed | 88.0 | 41.6 | 19.0 | 8.0 | 12.9 |
| RP+ReLU (RanPAC) | 95.2 | 48.8 | 23.1 | 9.7 | 15.7 |
| RanDumb (Ours) | 98.3 (+3.1) | 55.6 (+6.8) | 28.6 (+5.5) | 11.1 (+1.4) | 17.7 (+2.0) |

| Model | CIFAR100 |
|---|---|
| Joint | 79.58 |
| CNN x1 | 62.2 ±1.35 |
| CNN x2 | 66.3 ±1.12 |
| CNN x4 | 68.1 ±0.5 |
| CNN x8 | 69.9 ±0.62 |
| CNN x16 | **76.8 ±0.76** |
| ResNet-18 | 45.0 ±0.63 |
| ResNet-34 | 44.8 ±2.34 |
| ResNet-50 | 56.2 ±0.88 |
| ResNet-101 | 56.8 ±1.62 |
| WRN-10-2 | 50.5 ±2.65 |
| WRN-10-10 | 56.8 ±2.03 |
| WRN-16-2 | 44.6 ±2.81 |
| WRN-16-10 | 51.3 ±1.47 |
| WRN-28-2 | 46.6 ±2.27 |
| WRN-28-10 | 49.3 ±2.02 |
| ViT-512/1024 | 51.7 ±1.4 |
| ViT-1024/1546 | 60.4 ±1.56 |
| RanDumb (Ours) | 74.8 (-2.0) |

**Impact of Flip Augmentation.** We evaluate the impact of adding the flip augmentation on the performance of RanDumb in Table 6 (left, middle). Note that MNIST was not augmented. Augmentation provided large gains of 3.1% on CIFAR10, 1.7% on CIFAR100, and 0.4% on TinyImageNet. We did not augment the data further with RandomCrop transform as done with standard augmentations.

**Impact of Varying Ridge Parameter.** All prior experiments use a ridge parameter ($\lambda$) that increases with dataset complexity: $\lambda = 10^{-6}$ for MNIST, $10^{-5}$ for CIFAR10 and CIFAR100, and $10^{-4}$ for TinyImageNet and miniImageNet. Table 6 (left, middle) shows the effect of varying $\lambda$ on RanDumb's performance. With a smaller $\lambda = 10^{-6}$, CIFAR10, CIFAR100, TinyImageNet and miniImageNet all exhibit minor drops of 0.1%-1.7%, 0.8%, 0.8%. Increasing shrinkage to a $\lambda = 10^{-4}$ reduces CIFAR10 and CIFAR100 performance more substantially by 3% and 2.5% versus their optimal $\lambda = 10^{-5}$. On the other hand, this larger $\lambda$ leads to improvements of 0.5% and 1.8% on TinyImageNet and miniImageNet. This aligns with the trend that datasets with greater complexity benefit from more regularisation, with the optimal $\lambda$ balancing under- and over-regularisation effects.

**Comparison with Extreme Learning Machines.** We compared our random Fourier features with random projections based extreme learning machines, as recently adapted to continual learning by RP+ReLU [41] in Table 6 (left, bottom) with their best embedding size. Our method performs significantly better on each dataset, averaging a gain of 3.4%.

**Comparisons across Architectures.** In table 6 (right), we compare whether using random Fourier features as embeddings outperforms models across various architectures for continual representation learning. We use experience replay (ER) baseline in the task-incremental CIFAR100 setup (for details, see Mirzadeh et al. [45] as it differs significantly from earlier setups). Our comparison spanned various architectures. The findings revealed that RanDumb surpassed the performance of nearly all considered architectures, and achieved close to 94% of the joint multi-task performance. This suggests that RanDumb outperforms continual representation learning across architectures.

**Conclusion.** Overall, both random embedding and decorrelation are critical components in the performance of RanDumb. Using random Fourier features is substantially better than RanPAC. Lastly, one can substantially reduce the embedding dimension without a large drop in performance for large gains in computational cost, additional augmentation may further significantly help performance and optimal shrinkage parameter increases with dataset complexity. RanDumb outperforms continual representation learning across a wide range of architectures.

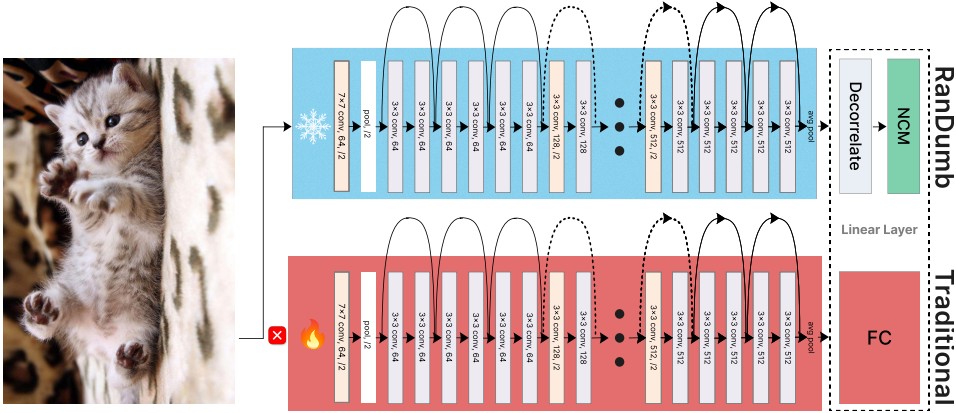

Figure 4: In previous experiments, models were trained from scratch, so we used random projection. Here, with a pretrained backbone, RanDumb starts with the frozen pretrained backbone to explore whether continual representation learning is necessary. By comparing this frozen backbone (RanDumb) with a continually trained one, we show that using the pretrained features consistently matches the best continually learned representations, similarly challenging the value of continual representation learning.

## 3.3 Should we learn representations when strong pre-trained features are available?

Say for a specific application (e.g., where the test data distribution is more or less known during training), practitioners should use strong pretrained models as a starting point as they are likely to perform better. However, we still ask the key question of whether representation learning is necessary by fixing the pretrained backbone and only training the linear classifier, as illustrated in Figure 4 in the next benchmark.

**Benchmark F.** We compare performance of approaches which do not further train the deep network like RanDumb against popular continual finetuning and prompt-tuning approaches in Table 7. We discover that prompt-tuning approaches completely collapse under large timesteps and approaches which do not finetune their pretrained model achieve strong performance, even under challenging one class per timestep constraint. Note that RanPAC [41] adds a RP+ReLU and finetunes in a first-session adaptation fashion over RanDumb, yet fails to achieve higher accuracies.

**Why is the performance of prompting methods so low?** We argue that the true test of continual learning lies in the ability to learn over prolonged periods. Our observations indicate that prompting methods collapse early across tasks because the learned prompts do not generalize effectively. Supporting this, parallel work [62] suggests that most current prompt-tuning methods lack prompt di-

Table 7: **Benchmark F** We compare RanDumb with prompt-tuning approaches using ViT-B/16 ImageNet-21K/1K pretrained models using 2 init classes and 1 class per task setting. Most prompt-tuning based methods collapse and RanDumb achieves either state-of-the-art or second-best performance. RanPAC-imp is an improved version of the RanPAC mitigating the instability issues identified in a previous version of this work.

| Method | CIFAR | IN-A | IN-R | CUB | VTAB |
|---|---|---|---|---|---|
| ViT-B/16 (IN-1K Pretrained) | | | | | |
| Finetune | 1.0 | 1.2 | 1.1 | 1.0 | 2.1 |
| L2P [68] | 2.4 | 0.3 | 0.8 | 1.4 | 1.3 |
| DualPrompt [67] | 2.3 | 0.3 | 0.8 | 0.9 | 4.2 |
| CODA-Prompt [59] | 2.6 | 0.3 | 0.8 | 1.9 | 6.3 |
| Adam-Adapt [81]) | 76.7 | 49.3 | 62.0 | 85.2 | 83.6 |
| Adam-SSF [81] | 76.0 | 47.3 | 64.2 | 85.6 | 84.2 |
| Adam-VPT [81] | 79.3 | 35.8 | 61.2 | 83.8 | 86.9 |
| Adam-FT [81] | 72.6 | 49.3 | 61.0 | 85.2 | 83.8 |
| Memo [80] | 69.8 | - | - | 81.4 | - |
| iCARL [54] | 72.4 | - | 35.2 | 72.4 | - |
| Foster [66] | 52.2 | - | **76.8** | 86.6 | - |
| NCM [31] | 78.3 | 44.3 | 62.5 | 84.8 | 88.2 |
| SLCA [79] | 86.3 | - | 52.8 | 84.7 | - |
| RanPAC [41] | **88.2** | 39.0 | 72.8 | 77.7 | 93.0 |
| RanPAC-imp [41] | 87.8 | 43.5 | 72.6 | **89.6** | 93.0 |
| RanDumb (Ours) | 84.5 | **49.5** | 66.9 | 88.0 | **93.6** |
| ViT-B/16 (IN-21K Pretrained) | | | | | |
| Finetune | 2.8 | 0.5 | 1.2 | 1.2 | 0.5 |
| Adam-Adapt [81] | 82.4 | 48.8 | 55.4 | 86.7 | 84.4 |
| Adam-SSF [81] | 82.7 | 46.0 | 59.7 | 86.2 | 84.9 |
| Adam-VPT [81] | 70.8 | 34.8 | 53.9 | 84.0 | 81.1 |
| Adam-FT [81] | 65.7 | 48.5 | 56.1 | 86.5 | 84.4 |
| Foster [66] | 87.3 | - | 5.1 | 86.9 | - |
| iCARL [54] | 71.6 | - | 35.1 | 71.6 | - |
| NCM [31] | 83.5 | 41.4 | 54.8 | 86.5 | 88.5 |
| SLCA [79] | 86.8 | - | 54.2 | 82.1 | - |
| RanPAC [41] | 89.6 | 26.8 | 67.3 | 87.2 | 88.2 |
| RanPAC-imp [41] | **89.4** | 33.8 | **69.4** | **89.6** | 91.9 |
| RanDumb (Ours) | 86.8 | **42.2** | 64.9 | 88.5 | **92.4** |

versity and can be characterized by a single prompt, making classification performance heavily reliant on the quality of that prompt. We hypothesize when prompts are designed for a large number of classes (e.g., 20 or 50), they produce generally discriminative representations that extend to future tasks. However, prompts designed for only two classes, as in our case, have limited discriminative power, leading to the collapse of prompt-tuning methods across tasks.

Overall, despite RanDumb being exemplar-free, it outperforms nearly all online continual learning methods across various tasks when exemplar storage is limited. We specifically benchmark on lower exemplar sizes to complement settings in which GDumb does not perform well.

# 4 Related Works

**Random Representations.** There have been extensive theoretical and empirical investigations into random representations in machine learning, compressed sensing, and other fields, often utilizing extreme learning machines [56, 14, 21] (see [30, 29] for a survey). Other investigations include efficient kernel methods using Fourier features and Nyström approximations [52, 70], and extensions to efficiently parameterize linear classifiers [2]. They are also embedded into deep networks [17, 35, 73, 16]. We tailored the already successful random fourier representations [52] to the problem at hand and applied to the online continual learning problem for the first time.

**Continual Representation Learning.** There are various works focusing on continual representation learning itself [53, 20, 39, 28], but they address the problem of alleviating the stability-plasticity dilemma in high-exemplar and offline continual learning scenarios where models are trained until convergence. In comparison, we focus on online and low-examplar regime.

**Representation Learning Free Methods in CL.** Several works have developed the idea of using fixed pretrained networks after adapting on the first task across various settings [50, 41, 24]. Our work contributes to this growing evidence, however, we do not perform first-task adaptation [47], and propose OAS-shrinked SLDA as structurally simplest but highly accurate continual linear classifier without any extra bells-and-whistles. Moreover, we are the first work to introduce a representation learning free method with random features for continually learning from scratch.

**Equivalent formulations to RanDumb.** If the classes are equiprobable, which is the case for most datasets here, nearest class mean classifier with the Mahalanobis distance metric is equivalent to linear discriminant analysis (LDA) classifier [42]. Hence, one could say RanDumb is exactly equivalent to a Streaming LDA classifier with an approximate RBF Kernel. Alternatively, one could think of the decorrelation operation as explicitly decorrelating the features with ZCA whitening [7].

# 5 Discussion and Concluding Remarks

Our investigation reveals a surprising result — simply using random embedding (RanDumb) consistently outperforms learned representations from methods specifically designed for online continual training. Furthermore, using random/pretrained features also recovers 70-90% of the gap to joint learning, leaving limited room for improvement in representation learning techniques on standard benchmarks. Overall, our investigation questions our understanding of how to effectively design and train models that require efficient continual representation learning, and necessitates a re-investigation of the widely explored problem formulation itself. We believe adoption of computationally bounded scenarios without memory constraints and corresponding benchmarks [51, 50, 22] could be a promising way forward.

**Limitations & Future Directions.** We currently do not provide theory or justification for why training dynamics of continual learning algorithms fails to effectively learn good representations; doing so would provide deeper insights into continual learning algorithms. Moreover, our proposed method, RanDumb with random Fourier features is limited in scope towards low-exemplar scenarios and online-continual learning. Extending studies on representation learning to high-exemplar and offline continual learning scenarios might be exciting directions to investigate.

**Social Impact.** RanDumb is an algorithm solely designed to perform a scientific study and we do not recommend use of RanDumb for any application in real-world production systems, hence no direct societal impact or explicit limitations on use in production systems is discussed.

## Acknowledgements

AP is funded by Meta AI Grant No. DFR05540. PT thanks the Royal Academy of Engineering. PT and PD thank FiveAI for their support. This work is supported in part by a UKRI grant: Turing AI Fellowship EP/W002981/1 and an EPSRC/MURI grant: EP/N019474/1. The authors would like to thank Arvindh Arun, Kalyan Ramakrishnan and Shashwat Goel for helpful feedback.

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

# A  Conceptual and Methodological Differences from GDumb [49]

Our core claim is that random representations from raw image pixels consistently outperform deep-learning-based representations designed for online continual learning. In contrast, GDumb's central claim is that continual learning methods need not actually minimize forgetting of previous online samples as their performance can be entirely recovered simply by a baseline using the latest memory.

The key differences between thee two works is as follows:

- **Forgetting of Online Samples (GDumb)**: GDumb argues that continual learning methods suffer from forgetting online samples and addresses this by relying entirely on memory. This dismisses the value of directly learning from online data, as GDumb does not retain or utilize information from online samples. RanDumb, in contrast, exclusively learns from online samples without using memory, emphasizing the significance of ongoing data streams for performance.

- **Inadequate Representation Learning (RanDumb)**: GDumb's approach to representation learning mirrors the experience replay (ER) baseline but does not address the quality of the learned representations. RanDumb explicitly focuses on the inadequacy of these representations and ablates their role to highlight their impact on continual learning. This reveals a key distinction in how each method evaluates representation quality in online learning settings.

Furthermore, the experimental setups for RanDumb and GDumb highlight their complementary nature:

- **Memory Settings**: RanDumb primarily targets low-memory environments, whereas GDumb excels in high-memory scenarios. For example, in rehearsal-free settings without exemplar storage, a common trend in continual learning, GDumb would inherently produce random performance due to its dependence on memory. RanDumb, on the other hand, thrives in these low-memory contexts, providing an alternative solution when memory is constrained.

- **Complementary Nature**: RanDumb and GDumb occupy complementary spaces within the continual learning landscape. RanDumb performs well in benchmarks where GDumb falters, and vice versa. As GDumb has been acknowledged as a valuable baseline, we argue that RanDumb similarly deserves recognition in the continual learning literature for its distinct strengths.

In summary, the only commonality between RanDumb and GDumb is that they serve as simple baselines. The points above underscore the fundamental distinctions between the two methods and the specific aims behind their development, as emphasized here and in the title of this work.

# B  Online Continual Learning: Our Setting

**Current Problem formulation.** We formally define the online continual learning (OCL) problem as follows. In classification settings, we aim to continually learn a function $f \colon \mathcal{X} \to \mathcal{Y}$, parameterized by $\theta_t$ at time $t$. OCL is an iterative process where each step consists of a learner receiving information and updating its model. For RanDumb, at each step $t$ of the interaction,

1. *One* data point $(x_t, y_t) \sim \pi_t$ sampled from a non-stationary distribution $\pi_t$ is revealed.
2. Learner updates the model $\theta_{t+1}$ using a compute budget, $B_t^{learn}$ and discards the datapoint.

**Simplifications by Compared Approaches.** Traditional online continual learning literature makes several concessions over this which makes the problem easier by allowing the datapoint to be saved for more timesteps. Training deep networks requires those simplifications as more data per batch helps stabilize the gradient updates. Typically, compared approaches store samples across for 10 timesteps, and performs an update with that batch of samples before discarding it. Most works further relax this by storing a memory buffer of samples indefinitely.

**Drawbacks.** Traditional online continual learning setups cannot effectively test for rapid adaptation because they use a class-incremental setup. Online learning is generally intended to enable quick adaptation to changing data and label distributions in a data stream. We believe a better formulation for online continual learning is described in [50].

