# OpenReview forum: "RanDumb: Random Representations Outperform Online Continually Learned Representations"
_NeurIPS.cc/2024/Conference — NeurIPS 2024 poster_

### Official Review · Reviewer_5ZCt · 2024-06-27

**Soundness:** 3
**Presentation:** 2
**Contribution:** 2
**Rating:** 5
**Confidence:** 3

**Summary:**

The paper shows that a fixed random feature space of the data, followed by some linear classifier, outperforms continually learned feature spaces across multiple benchmarks. The paper then shows an ablation study of the suggested method.

**Strengths:**

**Clarity:** The suggested idea is simple, and it is easy to understand how it works and why.

**Quality:** The random features are tested across multiple benchmarks, showing improvements in many of them. The ablation study is comprehensive. Overall, I find the results very convincing: I believe that RanDumb can outperform many continual learning algorithms, despite it being learned using a single pass over the data in a non-deep learning way.

**Significance:** This kind of meta-study, showing weak points of the entire field rather than weak points of a specific work is important for the community. The paper shows that continual learning methods are not the optimal solutions for many continual learning benchmarks.

**Weaknesses:**

I'll review this paper from 2 perspectives: the first is from a "technical" point of view, judging the actual suggested method RanDumb. The second is the "meta" point of view, judging the "meta" claim of the paper, that we should re-think CL methods, as they are not optimal for many of the existing CL benchmarks.

**Originality:**
From the technical point of view, the suggested method is simply an implementation of known methods on several different benchmarks, and as such has very limited originality. From the meta point of view, I think that the additional benefits of this paper over g-dumb [1] are somehow limited, as it has been shown before that CL methods do not optimally solve the CL benchmarks in many cases.

**Quality and significance:**
While the quality of the technical point is high, I'm not so convinced about the meta point. The discussion in the paper about the meta point is rather minimalistic and is left for the reader. A more thorough discussion will be useful, especially a more in-depth comparison with previous works that suggested similar things, such as g-dumb.
As I don't find the "technical" point of view of the paper novel significant enough to merit a top-tier publication in its own right, I think the paper should focus more on the meta point of view. Specifically, I find that there is a missing discussion on why the existence of simple methods that outperform CL methods on CL benchmarks should be so worrisome. The counter-argument is that the benchmarks themselves are more of "toy problems", and that achieving high performance on them is not the major goal of the works, but instead learning how to deal with problems that deep models encounter when facing changing data, including catastrophic forgetting and distribution shifts.

**Clarity:**
While the main idea is clear, the paper is very confusing to read. The different benchmarks are marked by letters, which makes it hard to follow which is which. Moreover, I often had to go to other papers to understand the experiments that were done in a better way, as this formulation was very confusing. This is a major problem in the current version, as a substantial part of the paper revolves around this formulation.




**Summary:**
I find the technical point of view good and convincing, but not significant and novel enough for to merit a publication in a top-tier conference. The meta point of view in my opinion is important and relevant for the community, but is not supported enough by the paper. Therefore, I am leaning toward rejection.










------

[1] Prabhu, Ameya, Philip HS Torr, and Puneet K. Dokania. "Gdumb: A simple approach that questions our progress in continual learning." Computer Vision–ECCV 2020: 16th European Conference, Glasgow, UK, August 23–28, 2020, Proceedings, Part II 16. Springer International Publishing, 2020.

**Questions:**

Can you please elaborate on the problems in CL that this paper introduces, and the difference between them and the points raised in g-dumb?

**Limitations:**

not relevant

---

> ### Author Rebuttal · Authors · 2024-08-06
>
> We thank the reviewer for their time and feedback. We appreciate the recognition that our idea is simple and easy to understand, as well as the recognition of the comprehensive nature of our ablation study and the improvements demonstrated across multiple benchmarks. The reviewer also pointed out that our meta-study, which highlights weak points of the entire field rather than specific works, is important for the community. We seek to address the concerns raised by the reviewer below. Below we have merged different parts of the review that come under the reviewer’s “meta-claim” and “technical claims” perspectives of the paper to provide a thorough reply in these aspects.
>
> ---
>
> ## Meta Claim
>
> > "... judging the 'meta' claim of the paper, that we should re-think CL methods, as they are not optimal for many of the existing CL benchmarks. I think that the additional benefits of this paper over g-dumb [1] are somehow limited, as it has been shown before that CL methods do not optimally solve the CL benchmarks in many cases. Can you please elaborate on the problems in CL that this paper introduces, and the difference between them and the points raised in g-dumb?"
>
> - Our meta-claim, as consistently stated across the work, is that *random representations of the raw pixels of the images consistently outperform the deep-learning based learned representations from methods specifically designed for online continual training*.
>
> - In contrast, the meta-claim made in GDumb is that continual learning methods forget all previous online samples, and their performance can be recovered by using the latest memory. Additionally, GDumb still uses deep models to justify their claims.
>
> ### Main differences between GDumb and RanDumb meta-claims
>
> - **GDumb primarily argues about forgetting of online samples:** GDumb relies entirely on memory and learns nothing from the online samples themselves. This ablates the effect of online samples on performance. In contrast, we only learn from online samples and use no memory.
>
> - **RanDumb primarily argues about inadequate representation learning:** GDumb learns representations similar to the experience replay (ER) baseline and does not make any claims regarding the quality of the representations learned via online representation learning. We specifically ablate representations and discuss the effects.
>
> Overall, there are critical fundamental differences between RanDumb and GDumb in their meta-points.
>
> ### Differences between the experimental settings of GDumb and RanDumb
>
> - **RanDumb focuses mainly on low-memory settings,** whereas GDumb primarily shines in high-memory settings. For example, it is trivial to note that in our primary rehearsal-free setting with no exemplars stored (currently popular in continual learning), GDumb would, by design, effectively provide random performance.
>
> - **RanDumb is entirely complementary to GDumb,** i.e., RanDumb shines in benchmarks where GDumb performs poorly and vice-versa. Therefore, just like GDumb, we believe that RanDumb would be a critical addition to the continual learning literature.
>
> Overall, we request the reviewer not to simply dismiss the meta-point because of a broad claim that both RanDumb and GDumb are simple baselines (with no other points in common), and we re-iterate that our meta-point is emphasized above and in the title.
>
> ---
>
> ## Technical Claim
>
> > "From the technical point of view, the suggested method is simply an implementation of known methods on several different benchmarks, and as such has very limited originality. I find the technical point of view good and convincing, but not significant and novel enough to merit a publication in a top-tier conference."
>
> We agree that a **part** of our work heavily relies on an existing and very well-cited work of Rahimi and Recht [52] to support our arguments. However, we believe that it is absolutely fine to do so for various reasons:
>
> - It is well accepted in the research community to use fundamental insights and findings from a related field. The above work is a fundamental work with thousands of citations of top-tier conference papers, which have been built on top of this work.
> - We are the first to show the effectiveness of the above approach in continual learning literature.
> - We had to combine various insights from the continual learning literature to make the above work effective across a variety of experiments we presented in this work.
>
> Just the fact that an approach seems simple should never be the reason to undermine its effectiveness, and we hope that the reviewer agrees with us.
>
> ---
>
> ## Presentation
>
> > "While the main idea is clear, the paper is very confusing to read. The different benchmarks are marked by letters, which makes it hard to follow which is which..."
>
> We would like to highlight that other reviewers found the presentation of the paper to be *"good"* or *“excellent.”* They found that *"This paper is well-written," "The motivation, method, and results are very clear and easy to follow,"* and even specifically, *"The paper is well-structured; it is relatively easy to browse the results and sections."*
>
> However, we do understand that it can sometimes be difficult to follow notations and references. Hence, we had summarized the important details about the different benchmarks relevant to our paper in a table alongside the “Benchmarks” paragraph in Section 3. We also describe critical aspects of the benchmarks in detail there.
>
> Having said the above, we do acknowledge that labeling experiments by alphabets might affect readability, and we will **improve this aspect by replacing these labels with short, explanatory phrases in our revised draft.**
>
> Could the reviewer suggest further ways to improve the structure of experiments? We are happy to make necessary changes.
>
> ---
>
> We hope we have addressed the major concerns of the reviewer, and are happy to answer any further questions/concerns. We look forward to a fruitful reviewer-author discussion phase.

---

> > ### Comment · Reviewer_5ZCt · 2024-08-11
> > **answer to the rebuttal**
> >
> > Thank you for the elaborate rebuttal.
> >
> > I still find many of the issues I've raised unanswered: I still find the discussion in the paper about the meta-point very minimalistic, and the paper's presentation very hard to follow.
> >
> > I still think that the paper is borderline, as it have both issues and merits. As there is no pure borderline score in this conference, I've put a borderline reject in my original review. Following the other reviewers and the rebuttal, I'll change my score to borderline accept.
> >
> > I hope that for future revisions of the paper, the authors will emphasize the discussion about the meta-point that appears in their rebuttals and implement the clarification to the suggested presentation.

---

> > > ### Author Response · Authors · 2024-08-13
> > >
> > > We would like to thank the reviewer again for the comments, and appreciate raising the score of our work.
> > >
> > > We will incorporate your comments in the paper to highlight the meta-point as you suggested. If you have any further questions or require clarifications then please let us know. We will be happy to address them.
> > >
> > > Thank you once again.

---

### Official Review · Reviewer_sAWs · 2024-07-08

**Soundness:** 3
**Presentation:** 4
**Contribution:** 3
**Rating:** 7
**Confidence:** 4

**Summary:**

This paper proposes RanDumb, a representation learning-free method for online continual learning (OCL). It uses data-independent random Fourier transform to project the data to a high-dimensional space (embed), scales the features to have unit variance (decorrelation), and finally performs classification with a nearest mean classifier (NCM) in the feature space.

The authors compares RanDumb with existing methods (when they are trained from scratch) on various OCL benchmarks. The authors additionally substitute the "embed" part of RanDumb with frozen pretrained representations and find that it still performs decently compared to existing methods that finetune the representations, on one of the benchmarks.

**Strengths:**

1. This paper is well-written and I did not find any major technical flaws.

2. I find that random representations outperform continually trained representations (from scratch) under the data/memory-constraint continual learning regime interesting and novel.

3. The benchmarks used are extensive, involving multiple datasets (MNIST, CIFAR, subsets of ImageNet, etc.), different levels of OCL constraints (e.g., number of classes per task), and different model architectures (ViT, ResNet, etc.).

**Weaknesses:**

1. I think the main novelty lies in the usage of random Fourier features because, without it, the proposed method is very similar to SLDA. Therefore, I'm not entirely convinced by the motivation. It makes sense that representation learning from scratch performs poorly in the low-data regime. In what cases do we not want to use a frozen pretrained representation directly?


2. The method involves using a very large embedding space (25K in most experiments), which raises concerns on efficiency. Kernel trick is efficient because you can compute the dot product in a high-dimensional space without explicitly projecting the data to that space, but I'm not sure how the dot product $\phi(x)^\top \bar{\mu}_i$ can be computed efficiently here. Could the authors elaborate on that?

**Questions:**

I listed my main concerns in the weaknesses section above and I only have some minor questions or comments here.

1. I wonder if it's worth providing the definition of OCL clearly so that readers not familiar with OCL can grasp the ideas more easily. Also, I recommend adding details on the method explicitly (even if it's just a few equations in the appendix) rather than only providing references. For example, I'd appreciate seeing how the inverse of the covariance matrix is computed online.

2. What part of the results are the reported numbers in prior work? Is it all the numbers except for the proposed method?

3. I thought Co$^2$L was a contrastive learning-based method. Why is it included in benchmark B.1 and not B.2?

4. Could the authors provide some hypothesis on why fine-tuning/prompt-tuning based methods collapse in Table 6?

**Limitations:**

The authors have addressed the limitations clearly in Sec. 5.

---

> ### Author Rebuttal · Authors · 2024-08-06
>
> We would like to thank the reviewer for their time reviewing our work and providing encouraging remarks such as *"no major technical flaws,"* *"interesting and novel,"* and *"extensive benchmarks."* We also appreciate giving the presentation of our work a remark of being *"excellent."*
>
> Below, we seek to address the potential weaknesses/questions raised by the reviewer in the same order as noted by the reviewer.
>
> ---
>
> > "I think the main novelty lies in the usage of random Fourier features because, without it, the proposed method is very similar to SLDA. Therefore, I'm not entirely convinced by the motivation. It makes sense that representation learning from scratch performs poorly in the low-data regime. In what cases do we not want to use a frozen pretrained representation directly?"
>
> When strong pre-trained features are available for a specific application (e.g., where the test data distribution is more or less known during training), practitioners should use them as they are likely to perform better. However, in the absence of such pre-trained representations, as is the case with online continual learning where the future datapoints/classes are unknown, we argue that a baseline, such as ours in this work that simply uses random features, should be first evaluated thoroughly before relying too much on the online representation learning methods that exist in the literature.
>
> We have made the above point empirically as well via extensive experiments where we show that several of the existing continual learning methods (including VR-MCL—the ICLR'24 best paper honorable mention [70]) fail to outperform our simple non-deep-learning based random feature baseline! We strongly believe that our random transform as a baseline will help in guiding future research directions. We are thankful to the reviewer for raising this question as we believe that adding a small discussion and related implications around this in our revised draft would strengthen our arguments further.
>
> ---
>
> > "The method involves using a very large embedding space (25K in most experiments), which raises concerns on efficiency. Kernel trick is efficient because you can compute the dot product in a high-dimensional space without explicitly projecting the data to that space, but I'm not sure how the dot product can be computed efficiently here."
>
> We will expand our discussion on implementation details of our method versus SLDA [27]. Briefly, computing the dot product was extremely fast. For our use case, the runtime (on an Alienware x17 R2 laptop for 25K dimensions) was **0.3±0.02 seconds**.
>
> However, as you asked for details in Q1, the major bottleneck is not the inner product, it is computing $S^{-1}$. We use a standard approach of solving linear systems $S \cdot x = \mu$ for this using a heavily optimized function in PyTorch called `torch.linalg.lstsq`. The solution $S^{-1} \mu$ is then used to compute $x^T (S^{-1} \mu) - \mu^T (S^{-1} \mu)$.
>
> Certainly, one could just invert the matrix $S$ at the beginning and then use the Sherman-Morrison formula for online updates as each update in our case is rank-1.
>
> ---
>
> > "I wonder if it's worth providing the definition of OCL clearly so that readers not familiar with OCL can grasp the ideas more easily. Also, I recommend adding details on the method explicitly (even if it's just a few equations in the appendix) rather than only providing references. For example, I'd appreciate seeing how the inverse of the covariance matrix is computed online."
>
> Thank you for this question. We agree that providing a more formal definition of OCL will improve the readability of the paper. We will provide that in our revised draft. For details about matrix inversion, please check the answer above.
>
> ---
>
> > "What part of the results are the reported numbers in prior work? Is it all the numbers except for the proposed method?"
>
> We clarify—all numbers in tables where the caption says (Ref: table and citation) except our method are taken from the aforementioned table in the cited paper. We ensure that the experimental settings are exactly the same. Please note that reporting numbers directly from prior works makes the baselines stronger as these are already heavily optimized by the authors of their corresponding papers, which most of the time is difficult to reproduce as the specific design choices (hyper-parameters, etc.) to achieve such numbers are not always well explained in the paper.
>
> ---
>
> > "I thought Co2L was a contrastive learning-based method. Why is it included in benchmark B.1 and not B.2?"
>
> Thank you for bringing this to our notice. We will correct this right away!
>
> ---
>
> > "Could the authors provide some hypothesis on why fine-tuning/prompt-tuning based methods collapse in Table 6?"
>
> Thank you for the great question. Based on recent research, and our observation that the collapse is early across tasks:
>
> Parallel work {1} (will be cited and included in the revised draft) suggests that most current prompt-tuning methods often suffer from a lack of prompt diversity and can be characterized with a single prompt. As a result, the effectiveness of classification depends heavily on the quality of that prompt.
>
> When designing a prompt for a large number of classes (e.g., 50 or 20), the prompt leads to generally discriminative representations (even for future tasks), whereas a prompt for just 2 classes, in our case, is limited in its discriminative power. This causes all prompt-tuning methods to collapse across tasks.
>
> {1} Thede et. al., Reflecting on the State of Rehearsal-free Continual Learning with Pretrained Models, CoLLAs 2024
>
> ---
>
> We hope we have addressed the major concerns of the reviewer, and are happy to answer any further questions/concerns. We look forward to a fruitful reviewer-author discussion phase.

---

> > ### Comment · Reviewer_sAWs · 2024-08-10
> > **Thanks for rebuttal**
> >
> > I thank the authors for addressing my concerns. I've raised my score to a 7.
> >
> > I hope the authors consider incorporating some of our discussions into the revisions. Additionally, I still don't like the use of the term "kernel **trick**" since the dot product is computed explicitly in the high-dimensional feature space. I also recommend clarifying early in the paper that the "representations" learned are on a very high dimensional space, since I think this is not very common when we use the word "representation learning" and could cause confusions.

---

> > > ### Author Response · Authors · 2024-08-13
> > >
> > > We would like to thank the reviewer again for an insightful review and valuable time. Appreciate it. We would also like to thank the reviewer for appreciating our rebuttal and raising the score. Thank you for that.
> > >
> > > > Additionally, I still don't like the use of the term "kernel trick" since the dot product is computed explicitly in the high-dimensional feature space.
> > >
> > > We agree with the reviewer and will fix this oversight on our part. As the reviewer rightly pointed out, since we directly compute the dot product in the high-dim space, we do not use the “trick” in line 77. We will correct this statement. Thank you for pointing this out.
> > >
> > > > I also recommend clarifying early in the paper that the "representations" learned are on a very high dimensional space, since I think this is not very common when we use the word "representation learning" and could cause confusions.
> > >
> > > We agree on this point too! We did highlight this in the introduction line 41, when introducing RanDumb design for the first time that projection is to a high dimensional space. However, we are happy to emphasise this earlier in the paper to avoid any potential confusion.
> > >
> > > Thank you once again for your time and insightful comments.

---

### Official Review · Reviewer_8BCQ · 2024-07-11

**Soundness:** 3
**Presentation:** 3
**Contribution:** 3
**Rating:** 6
**Confidence:** 5

**Summary:**

To obtain powerful representation in an online continual learning setting, the authors propose a new learning method referred to as RanDumb, that embeds raw pixels using a fixed random transform, approximating an RBF kernel initialized before any data is seen.

The proposed model trains a simple linear classifier on top without storing any exemplars, processing one sample at a time in an online continual learning setting.

Extensive experiments demonstrate its effectiveness and power with several ablations.

**Strengths:**

(+) Extending the investigation to popular exemplar-free scenarios with pre-trained models, this work shows that training only a linear classifier on top of pre-trained representations surpasses most continual fine-tuning and prompt-tuning strategies.

(+) The author's investigation challenges the prevailing assumptions about effective representation learning in online continual learning.

(+) The authors explain how the random Fourier basis (a low-rank data-independent approximation of the RBF Kernel) affects the representation of online continual learning along with the accuracy of RanDumb (embedding dimensions).

**Weaknesses:**

(-)  The structure of RamDumb (Fig. 1) is too simple to follow the structure. Could the authors depict the architecture of input, RBF-Kernel, pre-trained model (ViT), Decorrelate(D), and NCM(C)? or include an additional figure to illustrate the RamDumb.

**Questions:**

Please refer to the weakness.

**Limitations:**

The embedding (RBF-kernel) is unclear; the authors could better explain how the random Fourier projection affects continual learning input representation with simple math.

---

> ### Author Rebuttal · Authors · 2024-08-06
>
> We thank the reviewer for their time in reviewing our work and for providing their concerns and suggestions. We appreciate the recognition of the work's contribution in *"challenging the prevailing assumptions about effective representation learning in online continual learning."* The acknowledgment of our *"extensive experiments"* and results *"extending the investigation to popular exemplar-free scenarios with pre-trained models"* is highly valued.
>
> Below, we address the reviewer’s request for clarification on RanDumb’s structure.
>
> > P1 [Structure of RanDumb needs Clarity]: (i) Could the authors depict the architecture of input, RBF-Kernel, pre-trained model (ViT), Decorrelate(D), and NCM(C)? Or include an additional figure to illustrate the RanDumb structure.
> >
> > P2 [Embedding]: The embedding (RBF-kernel) is unclear; the authors could better explain how the random Fourier projection affects continual learning input representation with simple math.
>
> We would like to clarify the structure of RanDumb across benchmarks (and will add a figure in the revised draft to clarify this).
>
> - **For training from scratch**:
>   Raw flattened pixels are inputted, followed by random projection, decorrelation, and then classification using the NCM (as described in Equation 3). No input representations are learned; we use raw pixels as inputs, as illustrated in Figure 1.
>
> - **For pretrained models**:
>   The input image is passed through the pretrained model to obtain features, followed by decorrelation, and then NCM classification. Mathematically, this would be the same as Equation 3 if we overload $\phi$ as the pretrained model instead of random projection.
>
> We acknowledge our initial description was inadequate and appreciate you bringing this to our attention. We will clarify this in the paper with a separate figure for pretrained models and additional equations to avoid overloading $\phi$.
>
> We hope we have addressed the major concerns of the reviewer, and are happy to answer any further questions/concerns. We look forward to a fruitful reviewer-author discussion phase.

---

> > ### Comment · Reviewer_8BCQ · 2024-08-11
> >
> > Thank you for your kind response.
> >
> > I would maintain my score to see how the random Fourier projection (mechanism) affects continual learning or feature representations.

---

> > > ### Author Response · Authors · 2024-08-13
> > >
> > > We would like to thank the reviewer again for their time, feedback and are glad they liked our work.

---

### Official Review · Reviewer_k3sw · 2024-07-13

**Soundness:** 3
**Presentation:** 3
**Contribution:** 3
**Rating:** 7
**Confidence:** 3

**Summary:**

The authors show that a model with random Fourier features as a representation, followed by a normalisation then nearest class means for classification, outperforms most Continual Learning methods in online CL benchmarks. The representation is not learned as opposed to other online CL methods the model is compared against. Based on this result, the authors conclude that, in the online learning setting, continual learning algorithms fail to learn effective representations. The second main contribution is that this naive learning method bridges 70% to 90% of the performance of joint learning, leaving a small improvement gap to develop an effective continual representation learning method.

**Strengths:**

- Originality :
   - Even though the observation is simple, it highlights a blind spot in the field. Also, the naive model contrasts with most methods in the field and raises interesting questions.
- Quality :
   - The authors present extensive benchmarks and each benchmark is clearly motivated wrt the literature and the scope of the paper.
   - An ablation study on the components of the naive model is presented, highlighting that both the normalisation and Fourier features are critical for the performance.
- Clarity :
   - The motivation, method and results are very clear and easy to follow.
   - The paper is well structured as well, it's relatively easy to browse the results and sections.
- Significance :
   - This study will likely motivate subsequent works to investigate online representation learning

**Weaknesses:**

I don't see any major weaknesses, except the question I raise in the section below.

**Questions:**

One point I wanted to discuss was the following : One property of RanDumb is that training is not gradient based. This contrasts of all other training methods. Given that the other methods train the final layer with gradient based optimisation methods, it could be that the representations are still decent for previous tasks but that the forgetting occurs mostly in the final layer. I was thinking that, in order to validate that the other continual learning methods don't learn effective representations, one experiment could be : Run the full CL sequence on a given dataset with a given model, training the full model. then freeze the learned representation and check the joint accuracy with the frozen representation vs the joint accuracy with a fully trainable model.
I am curious about your thoughts on this point.

**Limitations:**

There are no potential negative societal impacts of the work. The authors adequately highlighted various limitations of their work. These limitations are out of scope of the current work and can be addressed in follow-up investigations.

---

> ### Author Rebuttal · Authors · 2024-08-06
>
> Thank you for your insightful question. To compare the effect, we will use two baselines: ER  [13] and iCARL [54].
>
> **Effect of gradient-free classifiers:** The primary difference between these methods is that ER trains the last layer along with the previous network, while iCARL learns a non-gradient based Nearest Class Mean (NCM) classifier. Similar to iCARL, RanDumb also employs an NCM classifier, but differs as it does not train a deep network.
>
>
> | Method          | Embedder      | Classifier | CIFAR10 (5/2) M = 0.1k | CIFAR10 (5/2) M = 0.2k | CIFAR100 (10/10) M = 0.5k | CIFAR100 (10/10) M = 1k | TinyImageNet  (100/2) M = 1k | TinyImageNet (100/2) M = 2k |
> |-----------------|---------------|------------|------------------------|------------------------|---------------------------|-------------------------|------------------------------|-----------------------------|
> | ER [13]         | Deep network  | Linear     | 19.4                            | 29.7                            | 8.7                                | 15.7                        | 2.5                             | 5.6                                |
> | iCARL [54]      | Deep network  | NCM        | 31.0                            | 33.9                            | 12.8                               | 16.5                        | 5.0                             | 6.6                                |
> | RanDumb (Ours)  | Random        | NCM        | 55.6                            | 55.6                            | 28.6                               |  28.6                       | 11.6                           | 11.6                                |
>
>
> Training the last layer in a gradient-based manner, as noted by the reviewer, can lead to an additional degree of forgetting. However, even after addressing this issue, there remains a significant gap  in performance that needs to be addressed, as highlighted by our comparison with RanDumb.
>
> The experiment proposed by the reviewer requires two passes over the data, not permissible in the OCL setting. We suggest that our experiment above might provide a valid comparison to clarify the concern while maintaining the online continual learning scenario.
>
> We hope we have addressed the major concerns of the reviewer, and are happy to answer any further questions/concerns. We look forward to a fruitful reviewer-author discussion phase.

---

> > ### Comment · Reviewer_k3sw · 2024-08-12
> > **Reply to rebuttal**
> >
> > Thanks to the authors for sharing the results !
> >
> > This experiment clarifies the concern I raised.
> > However I wanted to clarify that the experiment I suggested is a diagnostic rather than a method, therefore two passes over the data wouldn't be a concern. It would be the most direct comparison of the learned continual learning representations vs the random Fourier features, to validate whether they are strong enough.
> >
> > I will maintain my score, thanks !

---

> > > ### Author Response · Authors · 2024-08-13
> > >
> > > We would like to thank the reviewer again for their comments and for appreciating our work.
> > >
> > > > However I wanted to clarify that the experiment I suggested is a diagnostic rather than a method, therefore two passes over the data wouldn't be a concern. It would be the most direct comparison of the learned continual learning representations vs the random Fourier features, to validate whether they are strong enough.
> > >
> > > Yes, as a diagnostic tool two passes shouldn’t be a concern. We agree with that. We will investigate this further and, space permitting, we will make sure to include this diagnostic method in the revised draft as this might provide better insights.
> > >
> > >
> > > Thank you once again, appreciate it!

---

### Decision · Program_Chairs · 2024-09-25

**Decision:**

Accept (poster)

**Comment:**

The meta-message of this work is actually quite valuable to the community. Inline with the reviewers, I find this work to be ready for acceptance. I urge the authors to incorporate all the feedback they received in the camera ready version of the paper.